# A Diluted Electrolyte for Long-Life Sulfurized Polyacrylonitrile-Based Anode-Free Li-S Batteries

**DOI:** 10.3390/polym14163312

**Published:** 2022-08-15

**Authors:** Ting Ma, Xiuyun Ren, Liang Hu, Wanming Teng, Xiaohu Wang, Guanglei Wu, Jun Liu, Ding Nan, Baohua Li, Xiaoliang Yu

**Affiliations:** 1College of Chemistry and Chemical Engineering, Inner Mongolia University, Hohhot 010021, China; 2Inner Mongolia Key Laboratory of Graphite and Graphene for Energy Storage and Coating, School of Materials Science and Engineering, Inner Mongolia University of Technology, Hohhot 010051, China; 3College of Materials Science and Engineering, Qingdao University, Qingdao 266071, China; 4Department of Mechanical Engineering, Research Institute for Smart Energy, The Hong Kong Polytechnic University, Hong Kong, China; 5Rising Graphite Applied Technology Research Institute, Chinese Graphite Industrial Park-Xinghe, Ulanqab 013650, China; 6Shenzhen Key Laboratory on Power Battery Safety and Shenzhen Geim Graphene Center, Tsinghua Shenzhen International Graduate School (SIGS), Shenzhen 518071, China

**Keywords:** dilute electrolyte, Li-S battery, cycling stability

## Abstract

Lithium-metal batteries have attracted extensive research attention because of their high energy densities. Developing appropriate electrolytes compatible with lithium-metal anodes is of great significance to facilitate their practical application. Currently used electrolytes still face challenges of high production costs and unsatisfactory Coulombic efficiencies of lithium plating/stripping. In this research, we have developed a diluted electrolyte which is compatible with both lithium-metal anode and sulfurized polyacrylonitrile cathode. It presents a very high Li plating/stripping Coulombic efficiency of 99.3% over prolonged cycling, and the as-assembled anode-free Li-S battery maintains 71.5% of the initial specific capacity after 200 cycles at 0.1 A g^−1^. This work could shed light on designing a low-cost and high-performance liquid electrolyte for next-generation high-energy batteries.

## 1. Introduction

Lithium-ion batteries (LIBs) are widely used in portable electronic devices and electric vehicles due to advantages such as low cost, high performance, and environmental friendliness [1]. However, limited by the low theoretical capacity of traditional lithium-ion battery commercial cathodes (LiCoO_2_ [2], LiFePO_4_ [3], LiNi_x_Co_y_Mn_1-x-y_O_2_ [4]) and graphite anodes [5], the energy densities of current lithium-ion batteries struggle to meet the requirements of emerging industries such as electric vehicles [6]. Therefore, the development of energy storage systems with higher energy density is urgently required [7]. Metallic lithium possesses a high theoretical capacity (3865 mAh g^−1^) and a low electrode potential (−3.040 V versus standard hydrogen electrode) [8,9]. Therefore, lithium-metal batteries have become attractive candidates for next-generation batteries with high energy densities [10,11,12].

The practical application of lithium-metal batteries still faces two huge challenges. First, lithium metal has an ultra-high activity, which can irreversibly react with most electrolytes, consuming lithium ions and reducing Coulombic efficiency (CE) [13,14]. Second, lithium plating/stripping is uneven during the charging and discharging process [15,16], which leads to the growth of lithium dendrites, pierces the separator, and causes safety problems such as short circuiting of battery cells. To regulate Li deposition and suppress Li dendrite growth, researchers have developed various strategies, including current collector modification, electrolyte engineering, artificial solid electrolyte layer (SEI), and solid-state electrolytes. The modification of the current collector mainly reduces the lithium nucleation resistance by improving the affinity of the collector, and reduces current density and volume expansion by designing a 3D structure [17,18,19,20]. Electrolyte engineering mainly introduces some components of an electrolyte into SEI to improve certain physical and chemical properties of SEI [21,22]. Artificial SEI is an artificially designed protective film on Li metal with good properties identified by in/ex situ methods [23,24,25]. The solid electrolyte can reduce the risk of short circuit caused by lithium dendrite penetrating the separator, and thus greatly improve the safety of the battery [26,27].

The formation of robust SEI is a prerequisite for stable cycling and high CE of Li metal batteries [28,29]. The composition of SEI is closely related to that of the electrolyte [30,31,32]. Electrolyte engineering includes electrolyte additives [33,34,35], high-concentration electrolytes [36,37], and local high-concentration electrolytes [38,39,40]. Zhang et al. used fluoroethylene carbonate (FEC) as electrolyte additive to form LiF in SEI [41]. Zhang’s group achieved ultra-high CE (99.1%) with 4 M lithium bis(fluorosulfonyl)imide (LiTFSI) in 1,2-dimethoxyethane (DME) high-concentration electrolyte [42]. Although high-concentration electrolytes have achieved ultra-high CE values in lithium-metal batteries, the high viscosity, poor wettability, and high cost of high-concentration electrolytes hinder practical application. In order to solve the problem of high viscosity of high-concentration electrolyte, the concept of local high-concentration electrolyte was proposed, which used a non-solvated diluent to reduce the concentration of electrolyte [43]. For instance, Chen et al. designed 1.2 M lithium bis(fluorosulfonyl)imide in a mixture of flame-retardant triethyl phosphate/bis(2,2,2-trifluoroethyl) ether (1:3 by mol) localized high-concentration electrolyte, which led to a high CE of 99.2% in lithium-metal batteries [44]. However, the localized high-concentration electrolyte still had the disadvantages of high concentration and high cost, and CE still could not meet the actual application requirements in practical applications.

Lithium–sulfur (Li-S) batteries have a high theoretical capacity (1675 mAh g^−1^) and are an excellent choice for the next generation of lithium batteries [45]. However, the insulating properties of S and Li_2_S and the shuttle effect caused by the dissolution of intermediate polysulfides during cycling of Li-S batteries hinder the development of Li-S batteries [46]. Researchers composite S and C to form a cathode material (S@C) to enhance the conductivity of S and suppress the shuttle effect [47]. Composite cathodes prepared from a mixture of sulfur and pyrolytic polyacrylonitrile (S@pPAN) have attracted extensive research attention. Compared with S@C cathodes, S@pPAN can more effectively inhibit the dissolution of polysulfides, and enables good compatibility with carbonate electrolytes and little self-discharge [48]. However, S@pPAN did not perform well in ether-based electrolytes due to the slower reaction kinetics leading to partial dissolution of polysulfides [49]. Previously reported research works have introduced Se, an element of the same main group as S with a higher conductivity, into S@pPAN to improve the conversion reaction kinetics of the cathode [50,51].

In this work, we used lithium bisfluorosulfonimide (LiFSI) as lithium salt, 1,2-dimethoxyethane (DME) as solvent, lithium nitrate as additive, and the diluent 1,1,2,2-tetrafluoroethyl-2,2,3,3-tetrafluoropropylether (TTE) as a diluent, to prepare a diluted electrolyte, namely 0.67 M LiFSI in DME/TTE (1% LiNO_3_). A schematic illustration of the diluted electrolyte is shown in Figure 1a. The diluent has excellent chemical stability and will not change the original solvated structure of the lithium salt when added to the electrolyte. Therefore, diluting the electrolyte reduces the lithium salt concentration, thereby reducing viscosity and cost. Li-Cu half-cell using diluted electrolyte exhibited an ultra-high lithium plating/stripping CE of 99.3% at a current density of 0.5 mA cm^−2^. The diluted electrolyte also showed good compatibility with the pyrolyzed polyacrylonitrile and selenium disulfide composite cathode (pPAN/SeS_2_). The assembled anode-free lithium-sulfur battery delivered an ultra-high capacity-retention rate of 71.5% after 200 cycles. Therefore, this diluted electrolyte provides a new strategy for advancing the practical application of lithium-sulfur batteries.

## 2. Experimental

### 2.1. Synthesis of Materials

#### 2.1.1. Materials

Anhydrous lithium nitrate (LiNO_3_, 99.9% Alfa Aesar), (lithium bisfluorosulfonimide (LiFSI), 1,1,2,2-tetrafluoroethyl-2,2,3,3-tetrafluoropropylether (TTE), and 1,2-dimethyl ether oxyethane (DME)) were all purchased from TCI. Selenium disulfide (SeS_2_) and polyacrylonitrile (PAN) were both purchased from Sigma-Aldrich. Lithium foil (Li), aluminum foil (Al), copper foil (Cu), separator (Celgard 2400), carboxymethyl cellulose (CMC), styrene-butadiene rubber (SBR), and carbon black SUPER C45 (SP) were purchased from Shenzhen Branch Crystal Materials Technology Co., Ltd. (Shenzhen, China)

#### 2.1.2. Preparation of pPAN/SeS_2_

The PAN and SeS_2_ powders (1:4 *w*/*w*) were mixed and ground for 30 min before being sealed in a glass vessel and heat-treated at 380 °C for 8 h. The samples were then placed in a porcelain boat and kept at 350 °C for 6 h under a N_2_ protective atmosphere to remove excess sulfur and selenium elements, and then naturally cooled to obtain pPAN/SeS_2_ powder.

#### 2.1.3. Diluted Electrolyte Configuration

The purchased LiFSI was weighed and heated in a glove box at 60 °C for 4 h to remove traces of water. Solvent DME and diluent TTE were dewatered with molecular sieves for 48 h. Then, for electrolyte configuration, LiFSI and anhydrous lithium nitrate were first dissolved in DME, and stirred magnetically for 4 h. After the lithium salt was completely dissolved, TTE was added and stirred for 2 h.

### 2.2. Characterizations

The XRD patterns were collected using a Rigaku Smart Lab diffractometer (Japan) with Cu Kα radiation (λ = 1.5405 Å) in a scan range (2θ) of 10–80°. SEM images and EDS of pPAN/SeS_2_ powder were obtained by HITACHI-SU8220 (Japan) type field emission scanning electron microscope.

### 2.3. Electrochemical Measurements

CR2032 coin cells were assembled in a glove box filled with argon, with both O_2_ and H_2_O below 0.1 ppm. Cu (*φ*16 mm) and Li foil (*φ*15.5 mm) were applied to prepare Li-Cu half-cell, for evaluating Li plating/stripping efficiency. The cells were first cycled for 5 times at 50 μA in the voltage range of 0–1 V (vs Li^+^/Li) to remove copper foil surface impurities. Then, they were cycled at a current density of 0.5 mA cm^−2^ and a lithium deposition capacity of 1 mAh cm^−2^ for long cycling. In Li||pAN/SeS_2_ half-cell, the cathode is composed of a mixture of pPAN/SeS_2_, SP, and CMC/SBR in a mass ratio of 8:1:1, respectively, and the mass loading of the cathode is ~2 mg cm^−2^. Li||pPANSeS_2_ half-cells were cycled at a current density of 0.1 A g^−1^ in the voltage range of 1–3 V. The anode-free lithium-sulfur battery was assembled by first deeply lithiating the pPANSeS_2_ cathode through 2 mAh g^−1^, and then assembling it with copper foil to form a Cu||pPANSeS_2_ battery. The mass loading of the cathode is 4~6 mg cm^−2^. The Cu||pPANSeS_2_ cells were cycled at a current density of 0.1 A g^−1^ with a voltage interval of 1–3 V.

## 3. Results and Discussion

In this work, the effect of lithium salt concentration in the electrolyte on the electrochemical performance was first studied. Different concentrations of electrolytes nM LiFSI 1%wt LiNO_3_ in DME/TTE (1:2, *v*:*v*), (*n* = 0.33, 0.67, 1, 1.33) were configured and Li||Cu batteries were assembled to evaluate the compatibility of different electrolytes with lithium-metal anode. The result is shown in Figure 1a. The average CE of Li-Cu cells using a 0.33 M LiFSI in DME/TTE (1% LiNO_3_) electrolyte was 98.8% under test conditions of 0.5 mA cm^−2^ current density and 1 mAh cm^−2^ deposition amount, and the cells failed after 130 cycles. When the lithium salt concentration is increased to 0.67 M and 1 M, the average Coulombic efficiency is significantly increased to approximately 99.3% and the cycle life is also extended (about 200 cycles). With a continued increase of the electrolyte concentration to 1.33 M, the CE drops to 98.8%, and the CE value fluctuates significantly during the cycle, accompanied by a shortened cycle life. The batteries with 0.67 M and 1 M electrolytes showed similar CE and cycling stability. The CE value of Li plating/stripping in a certain electrolyte and cycling stability are mainly dependent on the component and microstructure of as-generated SEI. It can be deduced that 0.67 M and 1 M electrolytes benefit formation of SEI, with favorable properties such as good mechanical strength, uniform composition distribution, and smooth ion diffusion through SEI [52]. Considering the cost, the 0.67 M electrolyte was chosen as the optimal electrolyte. We further investigated the plating/stripping curves of Li-Cu cells with 0.67 M diluted electrolyte. The deposition overpotential of Li metal was 45 mV and remained stable during cycling (Figure 1c), further indicating that changing the diluted electrolyte could promote the stable electroplating/stripping of Li metal. We also tested the ionic conductivity and wettability of 0.67 M diluted electrolyte. As can be seen from Appendix A, it has a moderate ionic conductivity of 2.78 × 10^−4^ S cm^−1^ and also a good wetting capability in relation to copper foil, lithium foil, and the separator.

To reveal the effect of diluent TTE on the electrochemical properties of Li metal anodes, we prepared 2 M LiFSI in DME (1% LiNO_3_) and 0.67 M LiFSI in DME (1% LiNO_3_) electrolytes without diluent. The stable CE of the battery assembled with 2 M LiFSI in DME (1% LiNO_3_) electrolyte is about 98.5%. The CE fluctuates significantly after 100 cycles, and the battery fails at nearly 200 cycles. By contrast, the CE of the battery with diluted electrolyte 0.67 M LiFSI in DME (1% LiNO_3_) remained stable during 200 cycles (Figure 1d). We compared this electrochemical performance with those reported references in terms of electrolyte concentration, Coulombic efficiency, cycle number, and current density, as summarized in Table 1.

We further explored the effect of lithium nitrate on the electrochemical performance of the electrolyte. From the CE test results of the Li||Cu half-cell, the CE and cycling stability of the two electrolytes are generally similar in the initial 100 cycles (Figure 2a). Over prolonged cycling, the CE of the Li||Cu half-cell with 0.67 M LiFSI in DME/TTE (0% LiNO_3_) electrolyte fluctuates greatly, and the battery fails at the 130th cycle. The CE fluctuation over prolonged cycling mainly originates from increased internal resistances in the battery system. In contrast, the batteries with 0.67 M LiFSI in DME/TTE (1% LiNO_3_) electrolyte showed stable cycling. Next, we assembled Li||pPAN/SeS_2_ half-cells and tested the electrochemical performance of pPAN/SeS_2_ cathodes in the two electrolytes. Composition analysis and morphological characterization of pPAN/SeS_2_ were carried out. As can be seen in Appendix A, energy-dispersive X-ray spectroscopy (EDS) demonstrates uniform distribution of C, S, and Se elements, and the X-ray diffraction (XRD) pattern reveals an amorphous phase of pPAN/SeS_2_ (Appendix A). As shown in Figure 2b, the stability of the Li||pPAN/SeS_2_ half-cells assembled with the two electrolytes differs considerably at low active material loadings (the first discharge is not included in the figure, the first cycle in the figure corresponds to the second cycle). Specifically, the initial capacities of the two electrolytes are 541 mAh g^−1^ (0% LiNO_3_) and 540 mAh g^−1^ (1% LiNO_3_). The first 10 cycles show the same trend. As the cycle continued, the capacity of the electrolyte with 0% LiNO_3_ decreased significantly, and only 331 mAh g^−1^ remained after 200 cycles, corresponding to the capacity retention rate of 61.1%. As a comparison, the capacity of the diluted electrolyte with 1% LiNO_3_ remained stable during cycling, with a capacity of 535 mAh g^−1^ remaining after 200 cycles, and the capacity retention rate was as high as 99%. The galvanostatic charge–discharge curve results indicate that batteries applying the diluted electrolyte with 1% LiNO_3_ show almost no change in the charge–discharge plateau at 50th, 100th, and 200th, indicating a slight internal polarization (Figure 2c). On the contrary, with the progress of the cycle, the distance between charge and discharge platforms increases and the polarization is serious for the battery with 0% LiNO_3_ electrolyte, which is consistent with the cycling performance results.

For practical applications, we tested the cycling stability of Li||pPAN/SeS_2_ half-cells under high mass loading and lean electrolyte conditions, using a diluted electrolyte of 0.67 M LiFSI in DME/TTE (1% LiNO_3_). As can be seen in Figure 3a, an obvious capacity activation phenomenon can be observed. The second discharge capacity of the pPAN/SeS_2_ cathode is 489 mAh g^−1^, which increased slightly to 502 mAh g^−1^ after 50 cycles. After that, the specific capacity remained stable over prolonged cycling. The relatively lower specific capacity in the second cycle is because of the harsh testing conditions of lean electrolyte and high mass loading, which result in very difficult wetting of the electrode by the electrolyte. The active materials did not participate fully in the reaction during the initial few cycles. As the cycling progress continues, the pPAN/SeS_2_ cathode is activated and gradually wetted by the electrolyte. Thus, the specific capacity increases in the initial 15 cycles. After that, the quite stable cycling stability indicates the high lithium storage reversibility of the pPAN/SeS_2_ cathode in the diluted electrolyte. The galvanostatic current charge–discharge curves show that the charge–discharge voltage plateaus almost do not change over prolonged cycling, as shown in Figure 3b. It confirms that the internal resistance and thus voltage polarization of the Li-S battery cell is very small, which is consistent with the cycling performance results. The above results indicate that the diluted 0.67 M LiFSI in DME/TTE (1% LiNO_3_) electrolyte can enable good operation of Li-S cell even under harsh testing conditions of high cathode mass loading and lean electrolyte.

In order to maximize the energy density of the battery, we assembled an anode-free lithium-sulfur battery using the above diluted electrolyte. The pPAN/SeS_2_ cathode was first discharged to 0.28 V, followed by deep lithiation of Li-Al alloying reaction with a specific capacity of 2 mAh cm^−2^. After that, the lithiated cathode (pPAN/SeS_2_) was assembled with a bare Cu foil as the anode to form a Cu||pPAN/SeS_2_ anode-free cell. Figure 4a shows the cycling performance of the anode-free Li-S battery. It can be seen that the first discharge capacity of the Cu||pPAN/SeS_2_ full battery is 466 mAh g^−1^, which maintains 333 mAh g^−1^ after 200 cycles, with a capacity retention rate of 71.5%. The average CE is as high as 99.9% excluding the initial cycle. Here, we compare the electrochemical performances of our anode-free Li-S cell with those previously reported. As summarized in Table 2, we can see that our cycling stability is superior to those reported in other studies. It can be seen from the galvanostatic charge–discharge curves that the voltage platform only increases slightly with the cycle, indicating that the internal resistance of the anode-free cell is very small (Figure 4b).

## 4. Conclusions

In summary, a diluted electrolyte 0.67 M LiFSI in DME/TTE (1% LiNO_3_) was prepared for an advanced sulfurized polyacrylonitrile-based anode-free Li-S battery. At the anode side, the diluted electrolyte demonstrates a good compatibility with the lithium-metal anode, and thus it enables highly reversible Li plating/stripping with a high average CE of 99.3% over prolonged cycling. At the cathode side, it can effectively suppress dissolution and shuttling of intermediate polysulfides even under harsh testing conditions of high cathode mass loading and lean electrolyte. As a result, the fabricated Li-S battery maintains 71.5% capacity after 200 cycles at 0.1 A g^−1^, which is superior to those of previously reported anode-free Li-S cells. This work thus elucidates designing a low-cost diluted-liquid electrolyte towards next-generation long-life and high-energy batteries.

## Figures and Tables

**Figure 1 polymers-14-03312-f001:**
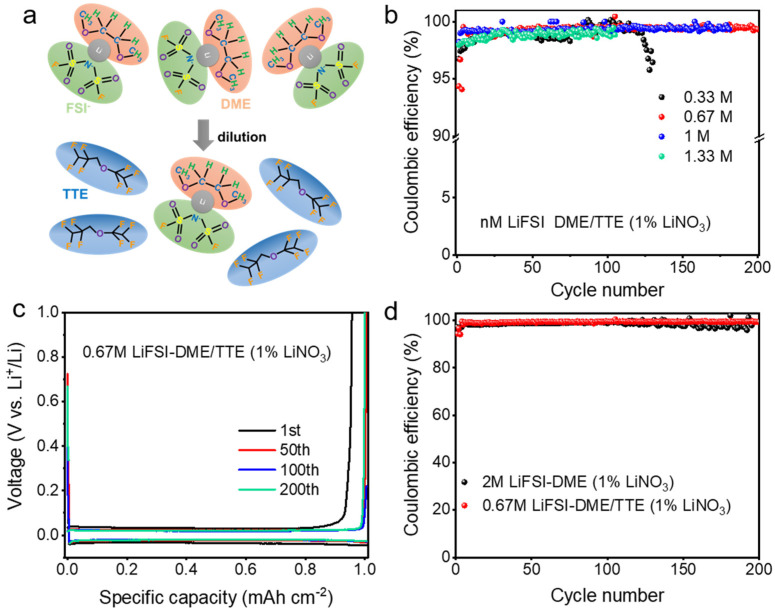
(**a**) Schematic illustration of the solvate structures of high–concentration electrolyte (top) and diluted electrolyte (bottom), with LiFSI, DME, and TTE as salt, solvent, and diluent, respectively; (**b**) CE test of Li||Cu half–cells electrolytes with different lithium salt concentration (0.5 mA cm^−2^, 1 mAh cm^−2^); (**c**) The galvanostatic charge–discharge curves of 0.67 M LiFSI in DME/TTE (1% LiNO_3_); (**d**) CE test of Li||Cu half–cells with different ratios of diluents.

**Figure 2 polymers-14-03312-f002:**
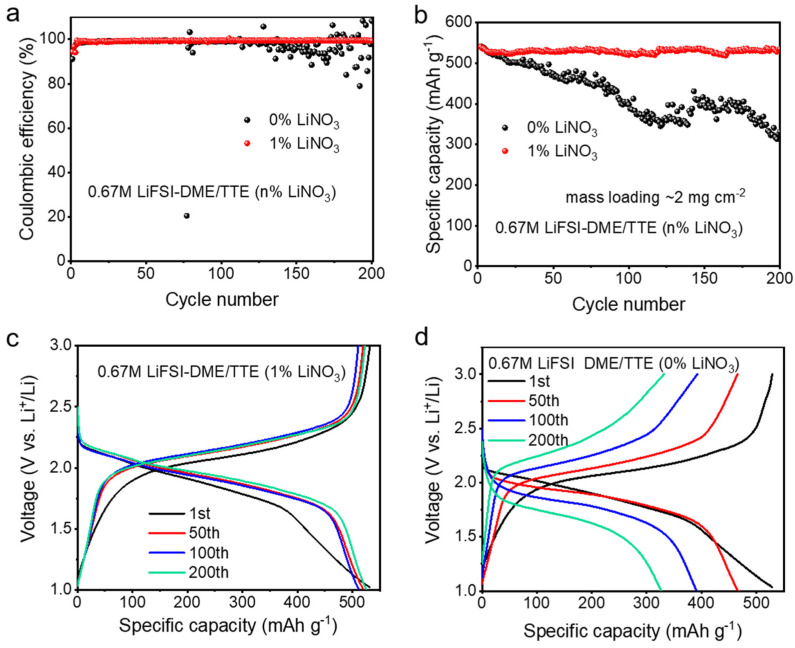
(**a**) CE testing of Li||Cu half–cells with diluted electrolytes of varying lithium nitrate content; (**b**) Cycling stability of Li||pPAN/SeS_2_ half–cells with diluted electrolytes of different lithium nitrate contents. The galvanostatic current charge–discharge curves; (**c**) 0.67 M LiFSI in DME/TTE (1% LiNO_3_); (**d**) 0.67 M LiFSI in DME/TTE (0% LiNO_3_).

**Figure 3 polymers-14-03312-f003:**
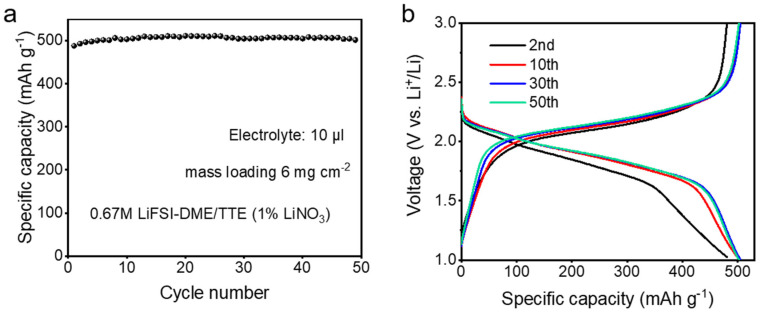
(**a**) Cycling stability; (**b**) Galvanostatic current charge–discharge curves of high mass loading and lean electrolyte Li||pPAN/SeS_2_ half−cells.

**Figure 4 polymers-14-03312-f004:**
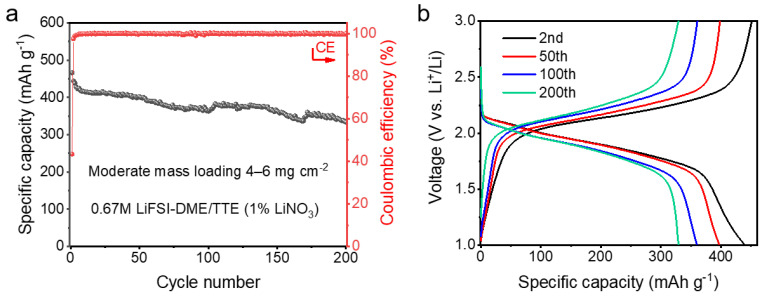
(**a**) Cycling stability; (**b**) Galvanostatic current charge–discharge curves of moderate mass loading and lean electrolyte Cu||pPAN/SeS_2_ half–cells.

**Table 1 polymers-14-03312-t001:** Electrochemical performance of LMBs in various electrolytes.

Electrolyte	Concentration	Li Plating/Stripping CE	Ref.
LiFSI-DME/TTE	0.67 M	99.3%, 200 cycles,0.5 mA cm^−2^, 1 mAh cm^−2^	This work
LiFSI-DME	4 M	98.5%, 300 cycles, 1 mA cm^−2^, 1 mAh cm^−2^	[42]
LiNO_3_ + LiFSI-DME	4 M	98.5%, 400 cycles, 0.5 mA cm^−2^, 0.5 mAh cm^−2^	[53]
LiTFSI-SL	1.3 M	96.6%, 100 cycles, 0.5 mA cm^−2^, 1 mAh cm^−2^	[54]
LiPF_6_ + LiDFOB- FEC/DMC/HFE	1.2 M	98%, 200 cycles, 1 mA cm^−2^, 1 mAh cm^−2^	[55]
LiFSI-DMC	10 M	99.2%, 200 cycles, 0.2 mA cm^−2^, 2.5 mAh cm^−2^	[56]

**Table 2 polymers-14-03312-t002:** Performance comparison of anode-free Li-S batteries between this and reported work.

Battery Chemistry	Cycling Stability	Loading	Ref.
Cu||pPAN/SeS_2_	71.5% retention over 200 cycles at 0.1 A g^−1^	4–6 mg cm^−2^	This work
ATCu||Li_2_S	64.8% retention over 120 cycles at 1.166 A g^−1^	4.5 mg cm^−2^	[57]
Cu||Li_2_S	70% retention over 100 cycles at 0.1166 A g^−1^	4 mg cm^−2^	[58]
Au/Cu||Li_2_S	69.5% retention over 150 cycles at 0.1 C	4 mg cm^−2^	[59]

## Data Availability

Not applicable.

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
