# Peer review of "A Diluted Electrolyte for Long-Life Sulfurized Polyacrylonitrile-Based Anode-Free Li-S Batteries"

_polymers, 2022, doi:10.3390/polym14163312_

Round 1

Reviewer 1 Report

In this study, the authors prepared a diluted electrolyte, 0.67M 74 LiFSI in DME/TTE (1% LiNO3) for enhancing the long life of Lithium metal batteries. The results is very promising for preparation of high performance liquid electrolytes. However, this research should be revised based on the following problems.

1. This study completely lacks characterizations of the DME/TTE (1% LiNO3) electrolyte. For example, viscosity, ionic conductivity (Nyquist plots).

2. Thermal stability of the DME/TTE (1% LiNO3) electrolyte should be investigated

3. The authors shoul provide a mechanism explain for the enhaced long life of the metal battery as schematic illustration

Reviewer 2 Report

The paper is devoted for new dilute electrolyte for long life sulfurized polyacrylonitrile-based anode-free Li-S batteries. The topic is generally interesting, however the paper contain unexplained places and need major revisions.

Line 133, ‘’The batteries with 0.67 M and 1 M electrolytes showed similar CE and cycling stability”, please explain why?

Please discuss more Figs. 3-4.

Why points are very scattered for black curves in Figs. 1d and 2a?

Please add more comparison of your obtained results with results presented in literature.

Conclusions should be rewritten in more informative way.

English need minor revisions.

Reviewer 3 Report

Authors present a study on the electrochemical performance of lithium-ion diluted electrolyte and its compatibility with both lithium metal anode and sulfurized polyacrylonitrile cathode.

The manuscript is concise and deals with a hot topic in the field of materials for lithium battery applications.

Although the manuscript deals mainly with the electrochemical characterization of the electrolytes in Li||pPAN/SeS2 and Cu||pPAN/SeS2 cells, in my opinion, it lacks a sufficient link with the field of polymer science.

The manuscript needs some revision before it can be considered for publication in Polymers, as detailed in the following queries list.

Q1) Introduction section

Authors omit some recent advances in lithium cathodes based on pyrolyzed polyacrylonitrile/selenium disulfide composite materials reported in the literature:

- Advanced Functional Materials 2014, 24(26), 4082–4089. doi:10.1002/adfm.201303909.

- Science Advances 2018, 4:eaat168. doi:10.1126/sciadv.aat1687.

- ACS Applied Materials & Interfaces 2019, 11, 29807-29813. doi:10.1021/acsami.9b07540.

- Sustainable Energy & Fuels 2020, 4, 3588-3596. doi:10.1039/d0se00512f.

Please, enrich the introduction section by providing a strong link with the role of pyrolyzed polyacrylonitrile in these cathode materials and its benefits compared to other carbon sources.

Q2) Methodology and results sections

Related to the previous query, can authors provide some physicochemical or structural characterization of their pyrolyzed polyacrylonitrile/selenium disulfide composite materials?

Q2) line 81

Related to the previous query, pPAN and SeS2 were not defined previously in the manuscript, please clearly define these terms.

Q3) line 113-114

Please correct the "pAN" term, it should be "pPAN" according to the previous definition.

Q4) Figure 2d

Please, modify the label of Figure 2d by substituting "1:2" for "0% LiNO3" to be consistent with the caption.

Round 2

Reviewer 1 Report

The current version can be accepted for publication

Reviewer 2 Report

Authors make proper corrections according to reviewer remarks and I suggest publish the paper as it is.

Reviewer 3 Report

Authors have addressed my queries and, in my opinion, the revised manuscript can be accepted for publication in Polymers.